# Effect of Different Land Use Types on the Taxonomic and Functional Diversity of Macroinvertebrates in an Urban Area of Northern China

Aoran Lu , Jiaxin Li, Biao Zheng and Xuwang Yin *

Liaoning Provincial Key Laboratory for Hydrobiology, College of Fisheries and Life Science,
Dalian Ocean University, Dalian 116000, China
* Correspondence: yinxuwang@dlou.edu.cn

**Abstract:** The urbanization of riverine landscapes is an increasing threat to river ecosystems. However, it is unclear which metrics can best assess the response of macroinvertebrates to the conversion of forested lands to urban and agricultural lands. The main goal of this study is to examine whether trait-based approaches are more sensitive than taxonomic approaches in distinguishing macroinvertebrate responses to different land use types in a highly urbanized area of northern China. Results based on 14 environmental variables showed a significant difference across a human-induced environmental gradient. The results showed that no significant differences were observed in terms of taxonomic diversity indices between the different land use types. Functional evenness (FEve) and Rao's quadratic entropy decreased with the increase in urban area caused by the intensification of human activity, demonstrating that functional diversity is more sensitive than taxonomic diversity in discriminating between different land use types. In addition, the results based on RLQ (physical–chemical variables (R), macroinvertebrate taxa (L), and species traits (Q)) and fourth-corner analyses indicated that the trait states of bi- or multivoltine, high dispersal capacity, and not-streamlined body shape were much higher in the agricultural area and positively related to farmland percentage. Taxa with large body size were dominant in urban areas and were positively correlated with EC. Overall, the observed responses of traits to environmental variables suggest that trait-based approaches should be incorporated into land use management for river restoration.

**Keywords:** macroinvertebrate; functional diversity; land use types; taxonomic diversity; urbanization



## 1. Introduction

The ecological impact of land use change on river and stream health is an important environmental issue worldwide Allan [1,2]. Increasing evidence suggests that changes in community composition caused by land use have been associated with the homogenization of taxa and the loss of taxonomic diversity [3]. For example, increased agricultural and urban land use in watersheds leads to the discharge of large amounts of domestic sewage and some industrial wastewater into urban rivers [4,5]. The abundance of pollution-tolerant taxa increases dramatically, and sensitive taxa are significantly reduced or disappear [6,7]. In addition, the expansion of agricultural land use within a watershed alters the nutrient dynamics due to the use of chemical fertilizers, and excess nitrogen and phosphorus inputs have modified community structure and biodiversity [8]. In this context, ecologists aim to understand the effects of land use change on biodiversity and explore which assessment methods can accurately examine the responses.

Macroinvertebrates are valuable and excellent indicators for biomonitoring and bioassessment [9]; compared to other aquatic indicators, macroinvertebrates have many advantages, such as sensitivity to stream disturbance, short life cycles, limited mobility, and ease of collection and identification [10]. Macroinvertebrates can also indicate habitat

degradation and fragmentation caused by land use change in the context of increasing concentrations of agricultural and urban pollutants [1,11,12].

The response of macroinvertebrates to the conversion of forest land to farmland and urban land is usually examined from the perspectives of taxonomic diversity [13]. For instance, changes in land use types can significantly alter community composition within a region by altering the physicochemical characteristics of the aquatic ecosystem [7]. The taxonomic diversity of macroinvertebrates significantly decreases along land use intensity and urbanization gradients [14]. However, taxonomy-based tools typically focus only on biodiversity measurements, ignoring the relationships between the environmental characteristics and the traits of macroinvertebrates in the ecosystem [15,16]. In aquatic ecosystems, trait-based methods have been confirmed to have certain advantages in assessing the impact of different land uses on macroinvertebrates [17–19]. The biological trait profile of communities offers an alternative approach to traditional measures of macroinvertebrate taxonomic identity and is less constrained by biogeographic influences [16]. Across broader regional extents and natural environmental gradients, the responses of certain functional traits to land use changes at various spatial scales were consistent because most stressors affect only certain trait categories [20]. Hence, in this context, assessing the response of functional traits to different land use types can not only distinguish multiple stressors but also provide early warning signals of ecosystem processes responses to such impacts before actual species loss [21–23].

This study aimed to assess the differences in taxonomic and functional diversity of macroinvertebrates in three land use types—forest land, farmland, and urban land—and the reasons for such differences during urbanization. Jinan, as a highly urbanized city, has gradually transformed most of its forest land into agricultural and urban land through disturbance from frequent human activities. The biodiversity of river ecosystems has decreased, and the functional composition has changed. Therefore, Jinan can serve as a typical city for studying the effects of land use change on urban river biodiversity and provide a theoretical basis for biodiversity conservation in urban rivers.

Three hypotheses were tested in our study. First, increasingly frequent human activities have significantly altered species composition in aquatic ecosystems. We hypothesized that species composition varies by land use type, with forested lands having a richer species composition and higher taxonomic diversity [24,25]. Second, some traits are more sensitive to land use change, and stressors associated with the shift from forest to urbanization and agricultural land can act as environmental filters that allow for differences in macroinvertebrate trait composition across land use types [26,27]. Third, trait-based approaches are more sensitive to environmental changes and can provide early warning signals of land use change before actual species loss [19,28]; therefore, rather than taxonomic diversity approaches, functional trait-based approaches can better distinguish differences in macroinvertebrates between land use types.

## 2. Materials and Methods

### 2.1. Study Area

This study was conducted in the regions of Jinan. Jinan is the capital of Shandong Province, located in the northern region of China. It has a short springs and autumns; hot and rainy summers; and long, cold winters. Based on the characteristics of land use types, the city of Jinan was categorized into mountain–hilly (Table S2), urban (Table S3), and agricultural land (Table S4) [29]. The southern area has a high degree of forest cover and is classified as a mountain–hilly area, and the primary flowing water system is the Yellow River. The middle region is more urbanized than the other areas and is therefore defined as an urban area, with the Xiaoqing River flowing through it. The northern region is a traditional agricultural area, and the Tuhai River, which is part of the Haihe River system, flows through the region as a large drainage river.

In 2015, the population of permanent residents in Jinan was 6.8 million, which consisted of 4.1 million individuals living in urban areas and 2.7 million individuals living

in rural areas. With the increase in population, land use and cover have dramatically changed [30] in this region over the past decades, reflecting the main types and degrees of anthropogenic disturbance in urban ecosystems. Repeat sampling was conducted at 44 sites during the spring, summer, and fall of each year from 2014–2016. These sites were divided into 15 mountain–hilly, 15 urban, and 14 agricultural sites (Figure 1).

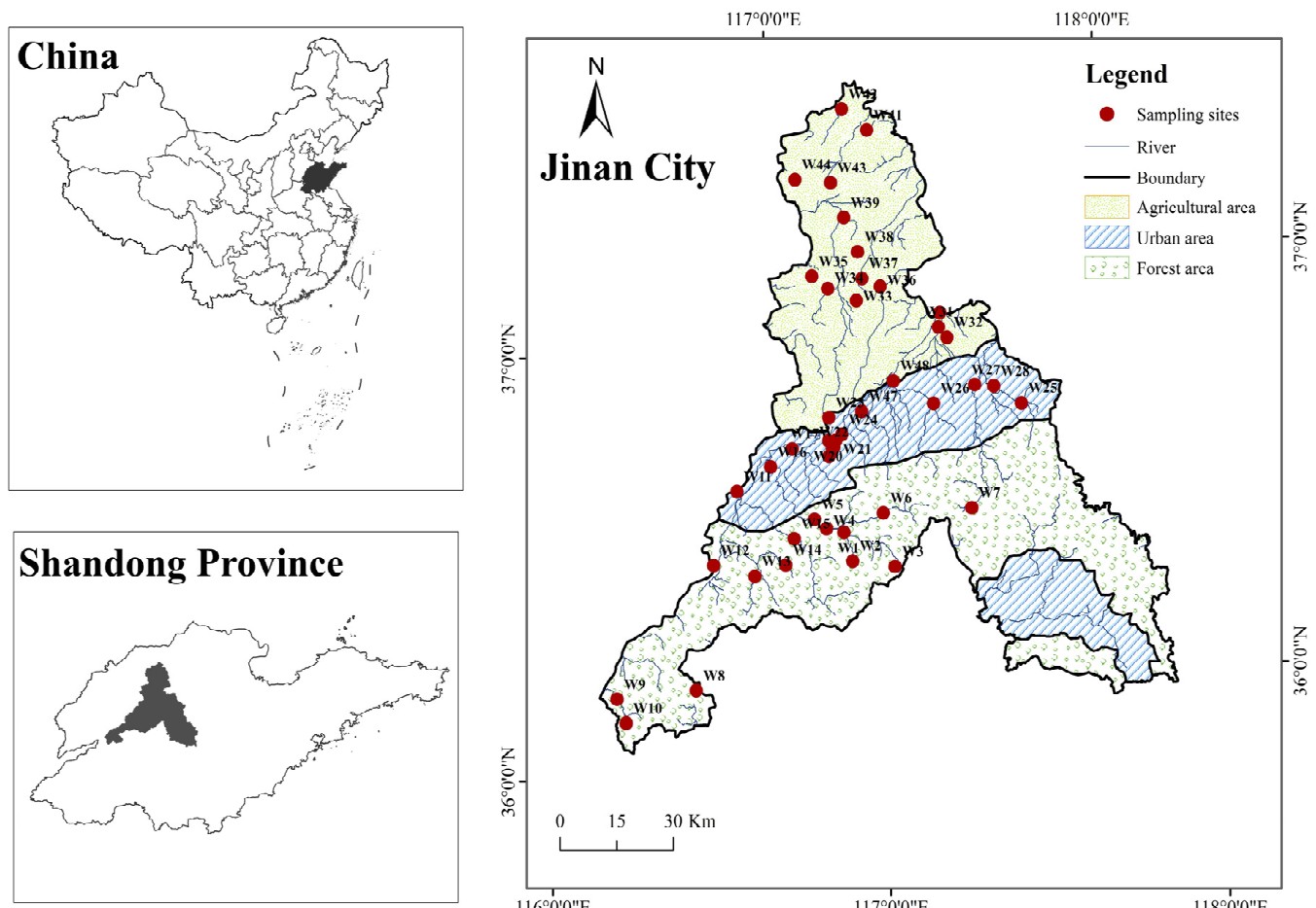

**Figure 1.** The distribution of 44 sampling sites in Jinan, Shandong Province, China.

### 2.2. Data Collection

Land use analysis was conducted using digital elevation models (DEM) with 30 m resolution in 2015, provided by the Chinese Academy of Sciences (http://www.cnic.cn/, accessed on 27 February 2022), to evaluate land use in the riparian zone around these sites. The Tabulate Area function in ArcGIS 10.5 (Esri, Inc, Redlands, CA, USA) was used to calculate the percentage of farmland, forestland, grassland, water area, and urban area within a 300 m buffer upstream of each sampling site to illustrate changes in the natural habitat structure of the river under different land use practices [31].

Eleven traits with thirty-six different trait categories were considered (Tables 1 and S5). Traits included those related to life history (voltinism and development), mobility (dispersal and drift), morphology (e.g., attachment, armoring, shape, respiration, and size), and ecology (rheophily, habit, and trophic habit). Trait categories were selected based on their mechanistic relationships with urban pollution. Trait availability and ease of measurement were taken into account during selection. There is considerable evidence in the literature that selected traits markedly respond to anthropogenic stress [32–37]. Information on trait categories identified was obtained mainly from the literature [38–40]. When trait information is not available at the genus or species taxonomic levels, we typically retrieve it at the family level instead. Affinity scores were used in a fuzzy coding approach that considers the

variability of macroinvertebrate life histories and better accounts for differences between species within a family and between species at different growth stages [41,42]. We use an affinity score range of 0–5 to describe the affinity of an organism for a trait attribute, with a score of 0 indicating that the organism has no affinity for a particular trait attribute and a score of 5 indicating a maximum affinity for that trait attribute. A score of 1 was used to indicate low affinity and a score of 3 to indicate moderate affinity [43]. Potential responses of macroinvertebrate community traits to land use type changes and associated stressors were predicted a priori.

**Table 1.** Macroinvertebrate traits and categories of used in the assessment of the trait–environment relationships across three different land use types in the present study.

| Traits | Category | Code | Reference |
|---|---|---|---|
| Voltinism | Semivoltine | Volt1 | [3,32,44] |
| | Univoltine | Volt2 | |
| | Bi- or multivoltine | Volt3 | |
| Development | Fast seasonal | Deve1 | [32] |
| | Slow seasonal | Deve2 | |
| | Nonseasonal | Deve3 | |
| Dispersal | No | Disp1 | [3,32] |
| | Low | Disp2 | |
| | High | Disp3 | |
| Drift | Rare | Drft1 | [3,32] |
| | Common | Drft2 | |
| | Very abundant | Drft3 | |
| Attachment | Free-ranging | Atch1 | [32] |
| | Sessile, sedentary | Atch2 | |
| | Both | Atch3 | |
| Armoring | None (soft-bodied forms) | Armr1 | [32] |
| | Poor (heavily sclerotized) | Armr2 | |
| | Good (e.g., some cased caddisflies) | Armr3 | |
| Shape | Streamlined (flat, fusiform) | Shap1 | [32,44] |
| | Not streamlined (cylindrical, round, or bluff) | Shap2 | |
| Respiration | Tegument | Resp1 | [3,32,44] |
| | Gills | Resp2 | |
| | Air (Plastron or spiracle) | Resp3 | |
| Size | Small (<9 mm) | Size1 | [3,32,44] |
| | Medium (9–16 mm) | Size2 | |
| | Large (>16 mm) | Size3 | |
| Rheophily | Depositional only | Rheo1 | [3,32] |
| | Depositional and erosional | Rheo2 | |
| | Erosional | Rheo3 | |
| Habit | Burrow | Habit1 | [3,32,44] |
| | Climb | Habit2 | |
| | Sprawl | Habit3 | |
| | Cling | Habit4 | |
| | Swim | Habit5 | |
| Trophic habit | Collector–gatherer | Trop1 | [3,32,44] |
| | Collector–filterer | Trop2 | |
| | Herbivore (scraper, piercer, and shedder) | Trop3 | |
| | Predator (piercer and engulfer) | Trop4 | |
| | Shredder (detritivore) | Trop5 | |

The measured physicochemical variables included water temperature, pH, electrical conductivity (EC), dissolved oxygen (DO) concentration, total nitrogen (TN), ammonium

nitrogen (NH$_3$-N), nitrate nitrogen, chemical oxygen demand (COD), biochemical oxygen demand (BOD), and total phosphorus (TP). Water temperature (Temp), DO, and EC were determined using a water quality analyzer (YSI85). After sampling, all water samples were placed in low-temperature incubators for storage and brought back to the laboratory. Physicochemical variables, such as COD, TN, NH$_3$-N, and TP, were measured following the Chinese surface water quality standards [45]. Macroinvertebrates were collected from 2014 to 2016 in the spring, autumn, and summer using a D-frame kick net (30 × 30 cm aperture) of mesh size 500 μm. Sampling was repeated at three locations within the same riffle. All samples were transferred through a 60-mesh sieve into 300 mL plastic bottles and stored in 90% alcohol solution. After the samples were brought back to the laboratory, the macroinvertebrates were separated from the impurities, such as rocks and debris. The benthic samples were transferred into 100 mL plastic bottles using the manual picking method, stored in 95% alcohol solution, and sorted and counted under a microscope or dissecting microscope. Macroinvertebrates were identified at the genus level or at the lowest practical taxonomic level in the laboratory using a compound light microscope and local guidelines for freshwater invertebrate identification [46–51].

### 2.3. Data Analysis

Taxonomic diversity across different land use types was compared using the Shannon diversity index, Pielou's diversity index, and species richness. Functional diversity analyses were conducted using the relative abundance of each trait category in the FD package to calculate the functional evenness (FEve), functional richness (FRic), functional divergence (FDiv), functional dispersion (FDis), and Rao's quadratic entropy (RaoQ) [52]. Functional richness (FRic) measures the size of the trait space occupied by the community [53]. Functional evenness (FEve) measures the homogeneity of the distribution of functional indices in the trait space, and functional divergence (FDiv) measures the proportion of species abundance at the edge of the trait space [54]. Functional dispersion (FDis) uses the relative abundance of species traits to calculate the dispersion of traits. Rao's quadratic entropy quantifies the divergent aspects of functional diversity as it not only takes into account differences in traits but also involves differences in abundance between species [55,56]. These indices can only reflect one aspect of functional diversity each, and combining them would provide a more accurate description [57].

R-environmental characteristics of sampling sites, L-species abundance across samples, Q-species traits (RLQ), and fourth-corner methods were used to identify the bivariate relationships between environmental variables and functional traits [37]. RLQ analysis maximizes the covariance between traits and environmental variables mediated by species abundance [16,58]. The fourth-corner approach quantifies and testes all correlations between trait categories and environmental variables. Three matrices were developed for the RLQ analysis, including a matrix of environmental variables, a matrix of functional traits, and a matrix of taxa abundance. Correspondence ordination analysis (CA) was performed on the taxa abundance matrices, principal component analysis (PCA) was applied to the functional trait matrices, and the Hill–Smith function was performed on the environmental variables [58]. The RLQ function was designed to integrate the results of the separate analyses performed on the R, L, and Q matrices. Monte Carlo tests were used to test the overall significance of Model 2 (H1: assumes no relationship between R and L) and Model 4 (H2: assumes no relationship between L and Q) [59]. According to the results of the RLQ ordination, the fourth-corner test was used to elucidate the relationship between functional traits and environmental variables to further confirm the ecological preferences of different land use types. The *p* values for the fourth-corner analysis were adjusted using the false discovery rate method.

The FD package was used to calculate the functional diversity indices [60], and the ade4 package was used for the RLQ and fourth-corner methods [61,62]. All statistical analyses were performed in R version 4.2.0 (R Development Core Team 2022). The Kruskal-Wallis test was performed using Origin 2022 (Origin Lab, Hampton, CA, USA).

### 3. Results

*3.1. Environmental Variables*

The results showed highly significant differences in physicochemical variables between the different land use types (Kruskal-Wallis test: $p < 0.05$, Table 2). Among these variables, the pH value, DO concentration, and percentage of forestland in mountain–hilly habitat were the highest. In contrast, the EC, ammonium nitrogen, TN, TP, COD, % farmland, and % urban land were markedly higher in the urban and agricultural areas than in mountain–hilly areas.

**Table 2.** Comparison of environmental variables (mean $\pm$ SD) in different land use type sites. Significant differences in environmental variables across different types of land use are indicated in bold (Kruskal-Wallis Test: $p < 0.05$). Mountain–hilly areas (15 sites); urban areas (15 sites); agricultural areas (14 sites).

| Variable | Mountain-Hilly Area | Urban Area | Agricultural Area | $p$ | $\chi^2$ |
|---|---|---|---|---|---|
| Water temperature (°C) | $21.57 \pm 4.89$ | $21.48 \pm 4.42$ | $20.94 \pm 4.88$ | 0.930 | 0.145 |
| pH | $8.24 \pm 0.37$ | $7.91 \pm 0.35$ | $8.17 \pm 0.33$ | **<0.001** | 33.132 |
| Electrical conductivity (EC) (s/cm) | $634.65 \pm 393.51$ | $1145.73 \pm 700.55$ | $1723.13 \pm 863.66$ | **<0.001** | 118.534 |
| Dissolved oxygen concentration (DO) (mg/L) | $8.15 \pm 1.52$ | $7.44 \pm 2.35$ | $7.49 \pm 2.45$ | 0.199 | 3.222 |
| Total nitrogen (TN) (mg/L) | $3.94 \pm 4.16$ | $5.86 \pm 4.49$ | $4.69 \pm 9.32$ | **<0.001** | 15.651 |
| Ammonium nitrogen ($NH_4$-N) (mg/L) | $0.96 \pm 3.11$ | $1.19 \pm 1.64$ | $2.05 \pm 8.78$ | **<0.001** | 11.931 |
| Nitrate nitrogen ($NO_3$-N) (mg/L) | $2.2 \pm 1.95$ | $3.81 \pm 3.73$ | $1.35 \pm 1.72$ | **<0.001** | 27.563 |
| Chemical oxygen demand (COD) (mg/L) | $20.56 \pm 17.89$ | $21.88 \pm 14.28$ | $32.18 \pm 31.45$ | **<0.001** | 27.522 |
| Biochemical oxygen demand (BOD) (mg/L) | $2.97 \pm 4.25$ | $3.58 \pm 3.76$ | $5.48 \pm 7.53$ | **<0.001** | 14.85 |
| Total phosphorus (TP) (mg/L) | $0.28 \pm 0.69$ | $0.33 \pm 0.58$ | $0.39 \pm 1.02$ | **<0.001** | 24.121 |
| Farmland (%) | $34.31 \pm 33.67$ | $24.52 \pm 24.33$ | $59.90 \pm 28.38$ | **<0.001** | 9.293 |
| Forest land (%) | $4.86 \pm 16.36$ | | | | 6.074 |
| Grassland (%) | $2.91 \pm 8.82$ | | | | 6.074 |
| Water area (%) | $24.50 \pm 24.57$ | $17.39 \pm 28.76$ | $12.35 \pm 23.55$ | 0.190 | 3.319 |
| Urban land (%) | $30.40 \pm 28.05$ | $58.03 \pm 41.09$ | $26.52 \pm 26.75$ | 0.120 | 4.233 |

*3.2. Taxonomic and Functional Diversity*

In total, 73 taxa were collected during the 2014–2016 sampling period, belonging to 12 orders, 25 families, and 30 genera. All taxonomic diversity indices used, namely species richness, the Shannon-Wiener index, Simpson diversity, and species evenness, did not differ markedly between different site groups (Figure 2).

Functional evenness (FEve) and Rao's quadratic entropy (RaoQ) were significantly higher in mountainous hilly areas than in agricultural areas (Figure 3). The functional diversity indices FRic, FDis, and FDiv exhibited no marked difference between the different land use types.

In terms of the functional traits, 23 trait categories showed significant differences across land use types (Kruskal-Wallis test: $p < 0.05$, Table 1, Figure 4, Table S1). Non-seasonal taxa dominated at agricultural sites, while fast-seasonal taxa showed high abundance at mountain–hilly sites. Regarding "dispersal", urban and agricultural areas had a much higher abundance of individuals with high dispersal ability. In terms of "drift", taxa with drifting ability were abundant at mountain–hilly sites. In addition, mountain–hilly areas exhibited a significantly higher abundance of taxa characterized by the traits "free-ranging" and "soft-bodied". In the case of "size" traits, small individuals predominated in mountain–hilly areas, whereas medium-sized taxa were most abundant at agricultural sites. In addition, the mountain–hilly area was dominated by burrowers, and the clingers were

most abundant in the agricultural area. Predators were most prevalent at the agricultural sites, while mountain–hilly areas had a higher abundance of collector–gatherers. No significant difference was observed between the three land use types for all trait categories of body shape, respiration, and rheophily.

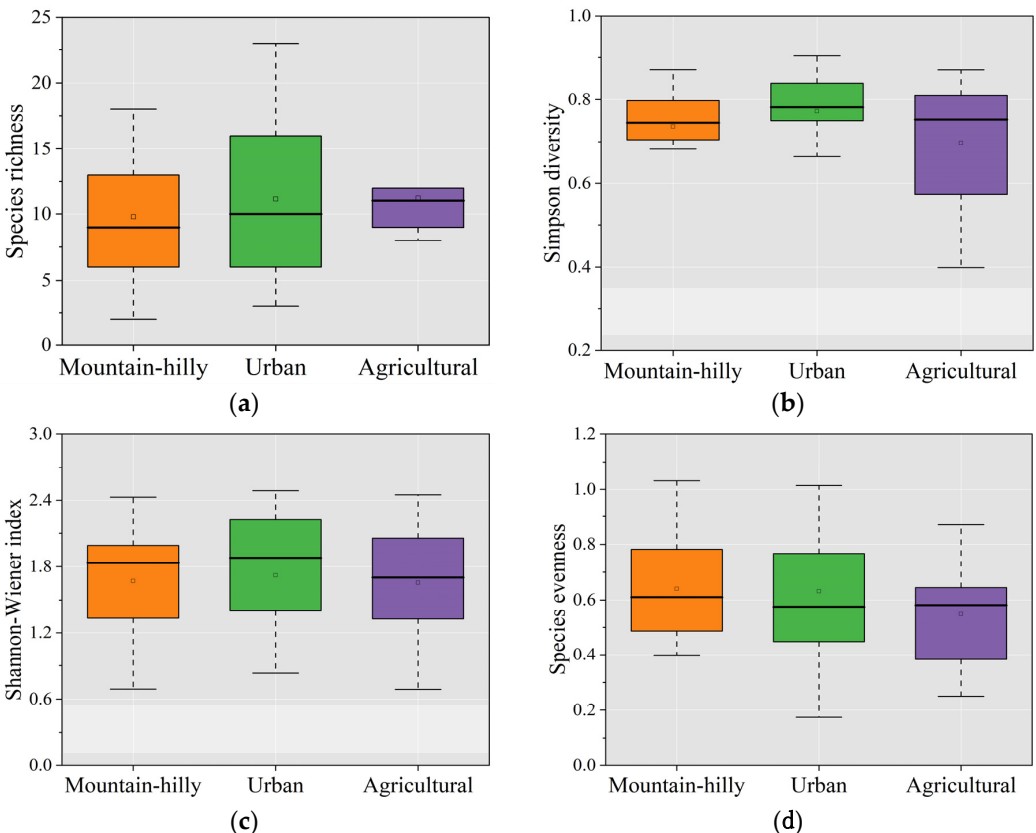

**Figure 2.** Box plots of taxonomic diversity indexes of richness, Simpson's diversity, Shannon–Wiener index, and evenness of macroinvertebrates distributed across different land use types. The lines in the boxes represent the median, the boxes represent the 1st and 3rd quartiles, and the lines at the top and bottom of the boxes represent the 5th and 95th percentiles, respectively. (**a**) Species richness in different land use types; (**b**) species diversity in different land use types; (**c**) species Shannon–Wiener index in different land use types; (**d**) species evenness in different land use types.

### 3.3. Relationship between Functional Diversity and Environmental Variables

The first two axes accounted for 91.9% (RLQ1 = 78.5%, RLQ2 = 13.4%) of the total variance between functional traits and environmental variables across the whole ecoregion (Table 3). The results of the fourth-corner analysis highlighted the bivariate association between environmental variables and functional traits (Figure 5). Among these variables, five trait categories showed predictive responses to environmental land use types, such as bi- or multivoltine, high dispersal capability, a rare occurrence of drift, not-streamlined body shape, and depositional habitat, which showed considerably positive correlations with the percent of farmland. The following traits of preference for free-ranging, soft-bodied forms and collector-gathering were positively correlated with total nitrogen. The composition of the five trait categories responded predictively with the TP concentration. For instance, organisms with a fast seasonal life cycle, a low dispersal capability, an abundant occurrence in drift, a small body size, and a burrowing habit were dominated by a high disturbance gradient with high phosphorus concentrations. Meanwhile, these traits were also negatively correlated with DO.

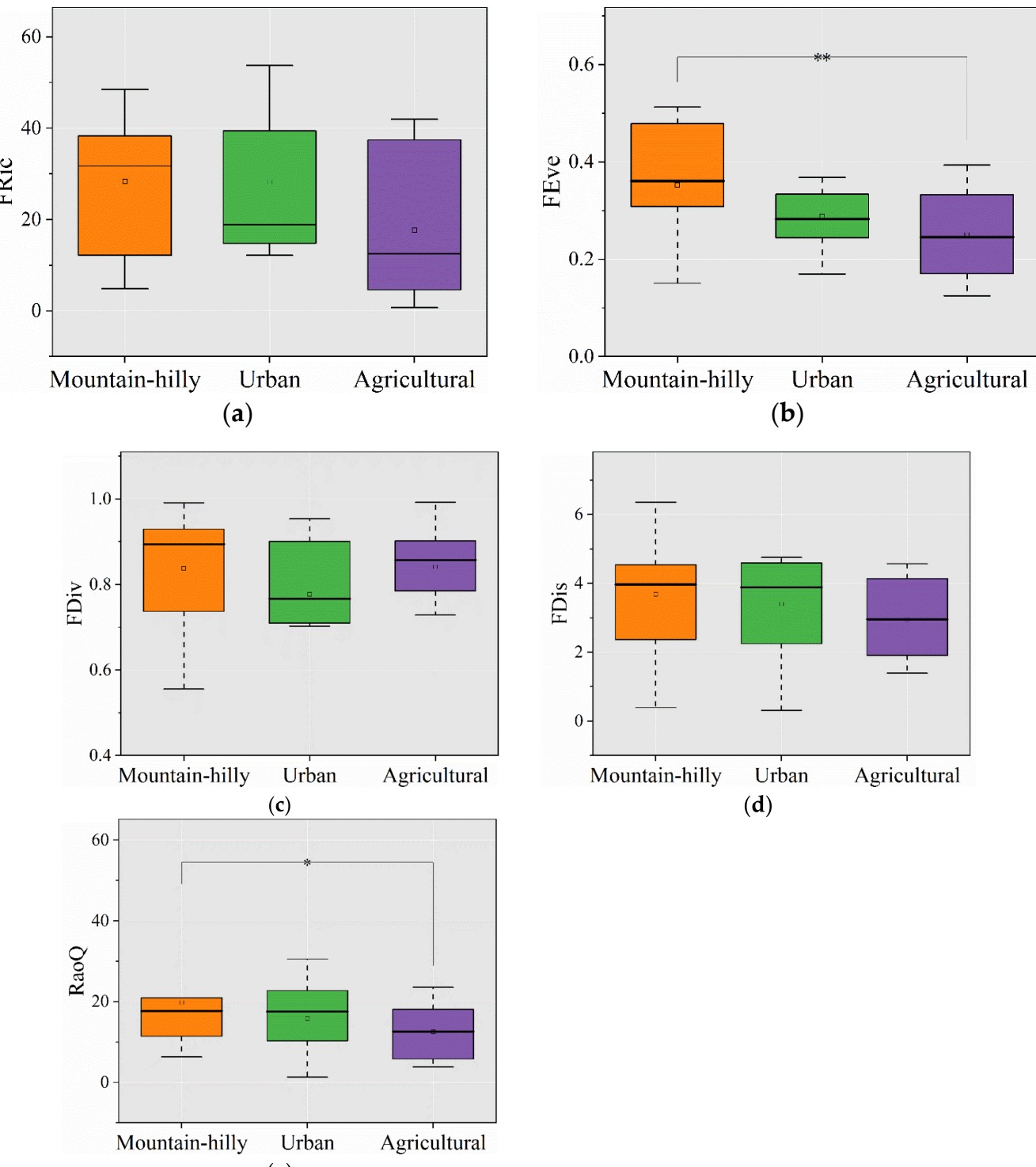

**Figure 3.** Box plots of functional diversity indexes of richness functional richness, functional evenness, functional divergence, functional dispersion, and Rao's quadratic entropy of macroinvertebrates distributed across different land use types. * significance level at 0.05 for non-parametric tests. ** significance level at 0.01 for non-parametric tests. (**a**) Functional richness (FRic) in different land use types; (**b**) functional evenness (FEve) in different land use types; (**c**) functional divergence (FDiv) in different land use types; (**d**) functional dispersion (FDis) in different land use types; (**e**) Rao's quadratic entropy (RaoQ) in different land use types.

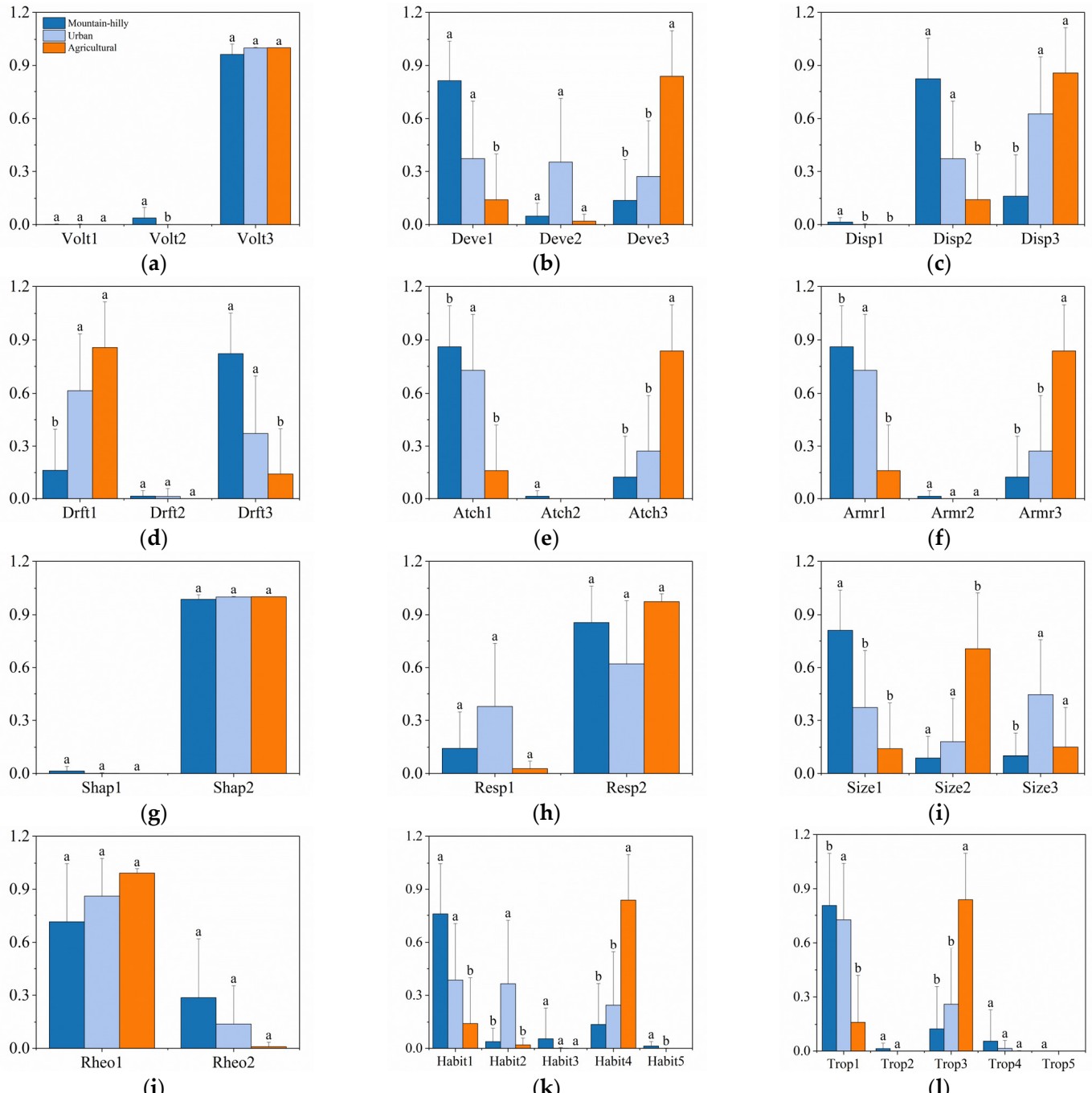

**Figure 4.** The structure of 36 categories of eleven traits at three land use types. Each bar represents the mean percentage with standard deviation of each trait category. (**a**) The trait of voltinism in different land use types; (**b**) the trait of development in different land use types; (**c**) the trait of dispersal in different land use types; (**d**) the trait of drift in different land use types; (**e**) the trait of attachment in different land use types; (**f**) the trait of armoring in different land use types; (**g**) the trait of shape in different land use types; (**h**) the trait of respiration in different land use types; (**i**) the trait of size in different land use types; (**j**) the trait of rheophily in different land use types; (**k**) the trait of habit in different land use types; (**l**) the trait of trophic habit in different land use types. Various letters denote the significant differences according to Kruskal-Wallis test at a 0.05 probability level.

**Table 3.** Results of the RLQ analysis. Eigenvalues and the first two RLQ axes account for the percentage of total inertia; covariation is the covariance between the two new sets of factor scores projected onto the first two RLQ axes; correlation is the correlation between the two new sets of factor scores projected onto the first two RLQ axes. Cumulative inertia is the variance of each set of factor scores; ratio is the variance of the first RLQ axis as a percentage of the first axis analyzed separately.

|  | **Axes1** | **Axes2** |
|---|---|---|
| Eigenvalue | 18.12 | 3.10 |
| % of total co-inertia | 78.47% | 13.43% |
| Covariation | 4.26 | 1.76 |
| Correlation | 0.66 | 0.46 |
| Cumulative inertia (environment) | 3.70 | 6.53 |
| Ratio (environment) | 84.61% | 91.26% |
| Cumulative inertia (traits) | 11.35 | 16.47 |
| Ratio (traits) | 99.08% | 83.98% |

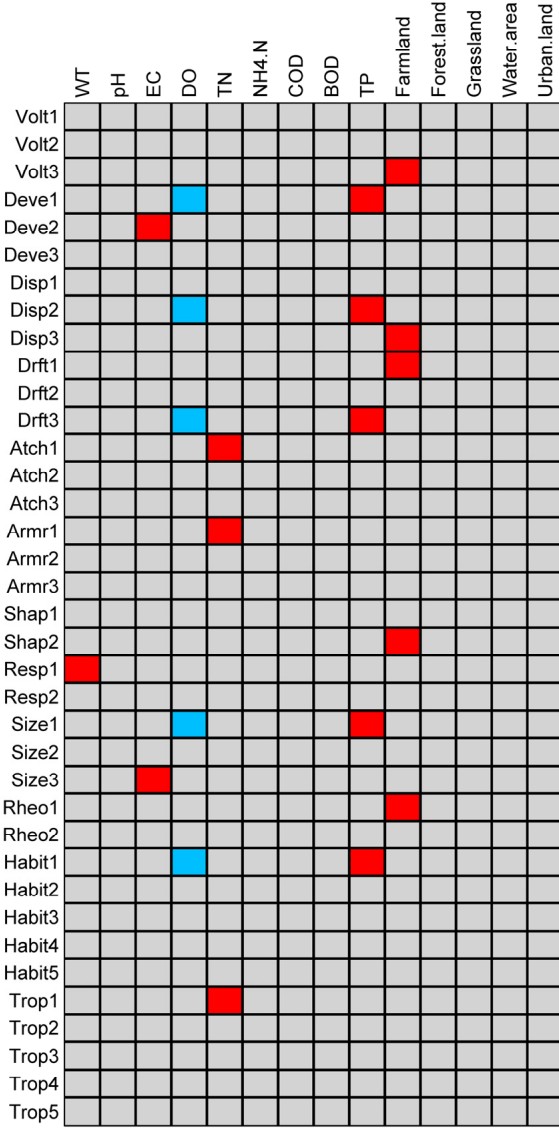

**Figure 5.** The summary of combining RLQ and fourth-corner analysis for macroinvertebrates traits and environmental variables. Red/blue cells indicate significant positive/negative relationships, respectively; grey cells indicate no significant relationships. The codes for macroinvertebrate traits and environmental variables are explained by Tables 1 and 2, respectively.

## 4. Discussion

Our results indicated that there were significant differences in the composition of macroinvertebrate traits between land use types in urban ecoregions. An interesting result showed that individuals characterized by traits related to resistance and resilience (i.e., small body size) occurred more frequently in mountain-hilly sites. The taxonomic diversity index in the present study was far less sensitive than the functional diversity index in distinguishing different land use types.

### 4.1. Response of Taxonomic Diversity to Different Types of Land Use

Inconsistent with previous studies [63–65], our results showed no significant differences in species composition and taxonomic diversity between land use types. The results indicated that taxonomic methods were not powerful tools for monitoring and evaluating the environmental impacts of land use changes on macroinvertebrates in highly urbanized ecoregions. This finding was in accordance with our third hypothesis that the taxonomic diversity index demonstrated low predictive capability for different land use types compared to the functional diversity index.

### 4.2. Response of Functional Diversity to Different Types of Land Use

As hypothesized, functional diversity showed a significant difference along the disturbance gradient, which could be a valuable indicator for distinguishing different land use types. Mountain–hilly areas exhibited higher FEve values than agricultural areas. Higher FEve values indicated that the distribution of trait abundance was relatively evenly disturbed in the functional space. Some researchers have reported relevant studies in which functional evenness (FEve) decreased with increasing disturbance, which was consistent with our findings [66,67]. RaoQ represents a mix between FRic and FDiv, which was demonstrated to be a promising indicator of disturbance [68]. The RaoQ values were significantly higher in mountain-hilly areas than in agricultural areas, indicating that land use changes caused by human activities, especially agricultural activities, lead to a decrease in water quality and habitat degradation in ecosystems [69]. Frequent anthropogenic stressors cause community variation and screening of biological trait composition, ultimately leading to changes in the functional diversity of aquatic insect communities.

In terms of the trait of large body size, we predicted that large body size would be less associated with highly disturbed sites and that small body size would be more prevalent as urban pollution increased. However, our results indicated the opposite trend [37]. The present study shows that taxa with large body size, such as gastropods, were dominant in urban areas. In addition, small-bodied species, such as chironomid species, were the most dominant taxa in mountain–hill areas. Our results support that those who reported large body size were associated with heavily impacted urban rivers [70–72]. In our study, electrical conductivity is an important variable influencing the pattern of body size distribution because it affects the osmotic balance. Macroinvertebrates are exposed to contaminants through external contact, so individuals with larger body size can minimize external exposure to dissolved salts by reducing the body surface area to volume ratio. Thus, high electrical conductivity in agricultural and urban areas contributed to the high relative abundance of the large-bodied species [19]. Small-bodied chironomids were associated with traits such as a fast seasonal life cycle, low dispersal ability, high drift occurrence, and burrowing. Previous studies have shown that these traits were highly resilient and resistant in the face of environmental disturbances, reducing the time to recover from disturbances [21]. This conclusion is supported by most studies on natural river disturbances [36,37,73]. However, in our study, rivers under different land use types were subjected to different levels of pollution from human activities, resulting in low species richness and mostly pollution-tolerant species. Therefore, our results do not support using these traits as indicators of urban river pollution levels. The trait states of lack of free-ranging, no body armor, and collector–gatherer feeding mode were associated with mountain–hilly sites and positively correlated with increasing TN. Burrowing species

preferred soft-bottom sediment habitats, thus coping with environmental pollution by using sediment as a buffer mechanism in water bodies [74]. The dominance of burrowers could be explained by the increased deposition of fine sediments in streambeds due to anthropogenic stressors in mountain–hilly areas, which reduces sediment heterogeneity [75]. The dispersal ability of aquatic dispersers in mountain–hilly areas was weak, which may be caused by the poor connectivity of aquatic systems and habitat fragmentation [76,77]. A higher proportion of gatherers was observed in the mountain–hilly areas and was significantly positively correlated with the TP and COD. Gatherers dominated the habitat of the mountain–hilly area, reflecting the fairly high levels of fine particulate organic matter in both water bodies and sediment [78]. The absolute predominance of gatherers may be partial because the main filterers are chironomids, which is inconsistent with the research conclusions of other scholars [79].

The percentage of farmland was positively correlated with a preference for bivoltine or multivoltine, high dispersal ability, a rare occurrence in drift, non-streamlined body shape, and depositional habitats. Previous studies have shown that taxa with traits of bi- or multivoltine mainly occur in disturbed sites because they have strong recolonization and refuge exploitation abilities in perturbed environments. For instance, studies [36] reported that the presence of organisms that exhibit bivoltine to pesticides impacted rivers, which was consistent with our research. The number of organisms with rare occurrences in drift increased with ongoing habitat destruction and degradation, mainly because the loss of boulders reduced habitat availability for taxa. The accumulation of sediments and the creation of depositional environments [80] for taxa prevented them from leaving stressed environments for more selective conditions. The trait of a non-streamlined body shape was also closely related to environmental stressors [81]. It has been reported previously that organisms with a streamlined body shape can reduce the drag effects caused by water currents. In the case of trophic habit, herbivores were highly abundant in agricultural areas, showing a predictable response to the anthropogenic disturbance gradient. The suspended particulate organic matter content was relatively high because of the large-scale use of fertilizer, herbicides, and pesticides, which contributed to the enhancement of aquatic plant growth, such as algae, and the increasing proportion of herbivores [70].

Respiratory traits indicated the response of the taxa to DO in the water column, which reflected DO concentration changes. Taxa breathing with gills were dominant under high DO conditions [82], while species with aerial respiration were dominant at sites with low DO and degraded ecosystem habitats. Our study found that the DO concentration was higher across the three land use types and that there was a higher abundance of taxa with gill respiration. The finding also supports the general idea that in high-DO environments, organisms with gill respiration are more common [83].

## 5. Conclusions

In conclusion, a comprehensive understanding of the differences in community characteristics and trait combinations under different land use types can contribute to the prediction of potential changes in aquatic community structure and ecosystem functioning in urban rivers. The results showed that the functional approaches of macroinvertebrates were better than taxonomic approaches for evaluating the ecological quality of urban river ecosystems. In addition, environmental variables played a dominant role in determining community trait composition. Therefore, this study has important implications for freshwater bioassessment in urban areas, especially in other areas with similar land use types.

**Supplementary Materials:** The following supporting information can be downloaded at: https: //www.mdpi.com/article/10.3390/w14233793/s1, Table S1. Kruskal-Wallis test results of the community-weighted means of trait values (CWM) of each trait category (mean ± SD) distributed across different types of land-use. Significant differences of environmental variables across different types of land-use sites are indicated in bold (Kruskal-Wallis Test: $p < 0.05$). Table S2. The proportions of land cover surrounding each site in mountain-hilly area. Table S3. The proportions of land cover surrounding each site in urban area. Table S4. The proportions of land cover surrounding each site

in agricultural area. Table S5. Species distribution under the three analyzed land use types. Trait abbreviations and descriptions of the are provided in Table 1.

**Author Contributions:** Conceptualization, X.Y. and A.L.; methodology, A.L.; software, A.L., J.L. and B.Z.; formal analysis, A.L. and J.L.; data curation, A.L.; writing—original draft preparation, A.L.; writing—review and editing, X.Y.; project administration, X.Y.; funding acquisition, X.Y. All authors have read and agreed to the published version of the manuscript.

**Funding:** This research was funded by the National Natural Science Foundation of China (41977193), and the National Science and Technology Basic Resources Survey Program of China (2019FY101700).

**Data Availability Statement:** Data are contained within the article or Supplementary Material.

**Acknowledgments:** Thanks are due to Jiaxin Li for assistance with the experiments and to Biao Zheng for valuable discussion.

**Conflicts of Interest:** The authors declare no conflict of interest.

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
