# Peer review of "Effect of Different Land Use Types on the Taxonomic and Functional Diversity of Macroinvertebrates in an Urban Area of Northern China"

_water, doi:10.3390/w14233793_

Round 1

Reviewer 1 Report

The manuscript is well written, but the material and methods are lacking in detail and there also seems to me to be a mix of results and discussion.

Author Response

Response

Dear reviewers:

Dear reviewers, thank you for your careful review and constructive suggestions regarding our manuscript. We have revised the manuscript in accordance with the comments and marked all the amends on our revised manuscript. Revised portion are marked in revised manuscript. The main corrections in the paper and the responses to the editor’s comments are as follows.

Answer to reviewer 1

Question1: “Please insert references”(It has a short springs and autumns, hot and rainy summers, and long, cold winters. Based on the characteristics of land-use types, the city of Jinan was categorized into mountain-hilly, urban, and agricultural land.)

Answer by authors to Q1: Following your comments, we have added relevant references. Please see page 2, line 90 in revised manuscript.

Question2:how do you do this??please explain in detail, how you acquired and where you acquired the satellite images, were they from the same year and period as the collection, how you performed the classification and in which program” (Satellite imagery)

Answer by authors to Q2: Following your comments, we have rewritten this paragraph to add the missing elements in the original paragraph. Please see page 3, lines 107-113 in revised manuscript.

Question3:I think that inserting in the table the reference from where you took the trait and what its functionality is important for the reader and makes the work even more elegant and complete” (Table 1.)

Answer by authors to Q3: Thank you very much for your comments, we have made changes to the previous table to add references to the table. We have also added a new table to make the data more clear. Please see page 4, lines 133-134 and pages 18 to 21, lines 374-377 in revised manuscript.

Question4: “Please explain why the use of all these indices are complementary?” (Functional diversity analyses were conducted using the relative abundance of each trait category in the FD package to calculate the functional evenness (FEve), functional richness (FRic), functional divergence (FDiv), functional dispersion (FDis), and Rao’s quadratic entropy (RaoQ))

Answer by authors to Q4: Following your comments, we referenced relevant references to describe each functional diversity index and further explain why these indices are complementary. Please see page 5, lines 158-166 in revised manuscript.

Question5: “reference” (Correspondence ordination analysis (CA) was performed on the taxa abundance matrices, principal component analysis (PCA) was applied to the functional trait matrices, and the Hill-smith function was performed on the environmental variables.)

Answer by authors to Q5: We have added the appropriate references. Please see page 5, lines 177 and 180 in revised manuscript.

Question6: “Version and year please” (R software.)

Answer by authors to Q6: We added the version information and year of the software. Please see page 5, line 187 in revised manuscript.

Question7: “Why this test was used” (Kruskal‒Wallis)

Answer by authors to Q7: The calculation of the differences in physicochemical parameters between different land use types were referred to in a paper from 2021. Please see Liu Z, Li Z, Castro DMP, Tan X, Jiang X, Meng X, Ge Y, Xie Z. Effects of different types of land-use on the taxonomic and functional diversity of benthic macroinvertebrates in a subtropical river network. Environ Sci Pollut Res Int. 2021 Aug;28(32):44339-44353. doi: 10.1007/s11356-021-13867-w. Epub 2021 Apr 13. PMID: 33847890.

Question8: “was in the objectives? why these results?” (The results showed highly significant differences in physicochemical variables between the different land-use types (Kruskal‒Wallis test: P <0.05, Table 2). Among these variables, the pH value, DO concentration, and percentage of forestland in mountain-hilly habitat were the highest. In contrast, the EC, ammonium nitrogen, TN, TP, COD, % farmland, and % urban land were markedly higher in the urban and agricultural areas than in mountain-hilly area.)

Answer by authors to Q8:  In our second hypothesis in this paper (Please see page 2, lines 77-80.), pressures associated with land use type shifts act as environmental filters and thus influence trait composition. Physicochemical parameters of water bodies affect trait composition, so we compared environmental factors between land use types to determine which traits could be used to indicate differences between land use types.

Question9: “Wouldn't that be discussion?” (Higher FEve values indicated that the distribution of trait abundance was relatively evenly disturbed in the functional space.)

Answer by authors to Q9: We apologize that we made the mistake. We have placed this sentence in the discussion section. Please see page 12, lines 290-291 in revised manuscript.

Question10: “I think there are results and discussion here” (Regarding “development”, the agricultural area was dominated by nonseasonal taxa, and fast seasonal taxa dominated the mountain-hilly area. Regarding “dispersal”, individuals with high dispersal capability occurred more frequently in urban and agricultural areas. In terms of “drift”, mountain-hilly sites have very abundant taxa with drifting ability. Mountain-hilly areas had more individuals with the trait categories “free-ranging” and “soft-bodied”. In the case of “size” traits, small individuals predominated in mountain-hilly areas, whereas agricultural areas had a much higher abundance of medium-sized taxa. In addition, the mountain-hilly area was dominated by burrowers, and the agricultural area had a significantly higher abundance of clingers. Herbivores occurred more frequently in agricultural regions than in mountain-hilly and urban regions, while mountain-hilly regions had a higher abundance of collector-gatherers. There was no significant difference among the three land-use types for all trait categories of body shape, respiration, and rheophily)

Answer by authors to Q10: Thank you for your comments, we have rewritten that section. Please see pages 8 to 9, lines 224-236 in revised manuscript.

Question11: “what is this?” (Error! Reference source not found)

Answer by authors to Q11: We sincerely apologize for our mistake, and the mistake has been corrected. Please see page 10, line 248 in revised manuscript.

Question12: “need to improve this figure” (Figure 5)

Answer by authors to Q12: Following your advice, we have improved this figure and adjusted the details. Please see page 11, lines 259-260 in revised manuscript.

Special thanks to you for your good comments.

Reviewer 2 Report

The list of species should be provided in Supplement with the data on distribution in three analyzed land-use types of areas with the codes of traits according to Table 1 for each species. For example, without this information it is not clear what type of "trophic habit" the large gastropods which dominate in urban lanscapes are assigned to?

In general, the taxonomic position of species integrates proposed functional traits, including body shape, trophical and respiration types etc. Analyzes of species richness in order or family rank possibly will give better picture than provided analyzes of total species richness or functional groups. I think it will be good task for future publication!

As for diversity estimation, I can recommend to use Renyi entropy as much more detail than separate indices. See, for example: Prokin A.A., Seleznev D.G. Structure of macrozoobenthos in floodplain lakes under different durations of spring flooding. Inland Water Biology. 2021. Vol.14, no 5. P. 573-580.

Author Response

Response

Dear reviewers:

Dear reviewers, thank you for your careful review and constructive suggestions regarding our manuscript. We have revised the manuscript in accordance with the comments and marked all the amends on our revised manuscript. Revised portions are marked in green in revised manuscript. The main corrections in the paper and the responses to the editor’s comments are as follows.

Answer to reviewer2:

Question 1: The list of species should be provided in Supplement with the data on distribution in three analyzed land-use types of areas with the codes of traits according to Table 1 for each species. For example, without this information it is not clear what type of "trophic habit" the large gastropods which dominate in urban lanscapes are assigned to?

Answer by authors to Q1:  We appreciate your opinion and thank you so much. We have added a table according to your comments (Table S5). Please see pages 18 to 21, lines 374-377 in revised manuscript.

Question 2: In general, the taxonomic position of species integrates proposed functional traits, including body shape, trophical and respiration types etc. Analyzes of species richness in order or family rank possibly will give better picture than provided analyzes of total species richness or functional groups. I think it will be good task for future publication!

Answer by authors to Q2: According to your opinion, we have analyzed the species richness in order and family among different land use types separately. In future studies, we will focus on the issue you pointed out. In this study, species richness calculated using species density was consistent with the diversity index, and we discussed and decided to use the previous calculation method. Thank you again for your comments. Please see page 7, lines 205-206 in revised manuscript (The PDF file contains two images).

Question 3: As for diversity estimation, I can recommend to use Renyi entropy as much more detail than separate indices. See, for example: Prokin A.A., Seleznev D.G. Structure of macrozoobenthos in floodplain lakes under different durations of spring flooding. Inland Water Biology. 2021. Vol.14, no 5. P. 573-580.

Answer by authors to Q3: We have downloaded the paper and read it carefully. Unfortunately, our knowledge base in statistics and physics is weak. After consulting with people familiar with statistics and physics, our weak knowledge base still does not allow us to grasp the meaning of the “Renyi entropy” and calculations. We are very sorry that there is no way we can get around this problem.

Special thanks to you for your good comments.
